# What if Neural Networks had SVDs?

**Alexander Mathiasen**[*]  **Frederik Hvilshøj**[*]  **Jakob Rødsgaard Jørgensen**[*]

**Anshul Nasery**[* †]  **Davide Mottin**[*]

## Abstract

Various Neural Networks employ time-consuming matrix operations like matrix inversion. Many such matrix operations are faster to compute given the Singular Value Decomposition (SVD). Techniques from [10, 17] allow using the SVD in Neural Networks without computing it. In theory, the techniques can speed up matrix operations, however, in practice, they are not fast enough. We present an algorithm that is fast enough to speed up several matrix operations. The algorithm increases the degree of parallelism of an underlying matrix multiplication $H \cdot X$ where $H$ is an orthogonal matrix represented by a product of Householder matrices.

## 1   Introduction

What could be done if the Singular Value Decomposition (SVD) of the weights in a Neural Network was given? Time-consuming matrix operations, such as matrix inversion [6], could be computed faster, reducing training time. However, on $d \times d$ weight matrices it takes $O(d^3)$ time to compute the SVD, which is not faster than computing the matrix inverse in $O(d^3)$ time. In Neural Networks, one can circumvent the SVD computation by using the SVD reparameterization from [17], which, in theory, reduces the time complexity of matrix inversion from $O(d^3)$ to $O(d^2)$. However, in practice, the SVD reparameterization attains no speed-up for matrix inversion on GPUs.

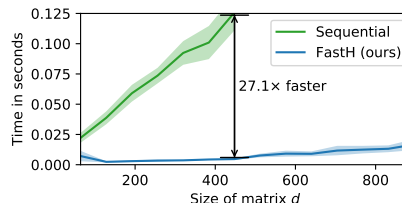

Figure 1: Time consumption of matrix inversion in Neural Networks. The plot compares FastH against the sequential algorithm from [17] (see Section 4).

The difference between theory and practice occurs because the previous technique increases sequential work, which is not taken into account by the time complexity analysis. On a $d \times d$ weight matrix, the previous technique entails the computation of $O(d)$ sequential inner products, which is ill-fit for parallel hardware like a GPU because the GPU cannot utilize all its cores. For example, if a GPU has 4000 cores and computes sequential inner products on 100-dimensional vectors, it can only utilize 100 cores simultaneously, leaving the remaining 3900 cores to run idle.

We introduce a novel algorithm, FastH, which increases core utilization, leaving less cores to run idle. This is accomplished by increasing the degree of parallelization of an underlying matrix multiplication $H \cdot X$ where $H$ is an orthogonal matrix represented by a product of Householder matrices. FastH retains the same desirable time complexity as the sequential algorithm from [17] while reducing the number of sequential operations. On a mini-batch of size $m > 1$, FastH performs $O(d/m + m)$ sequential matrix-matrix operations instead of $O(d)$ sequential vector-vector operations.

In practice, FastH is faster than all algorithms from [17], e.g., FastH is 27 times faster than their sequential algorithm, see Figure 1. Code www.github.com/AlexanderMath/fasth.

---
[*]Aarhus University, {alexander.mathiasen, fhvilshoj, mrjakobdk}@gmail.com, davide@cs.au.dk
[†]Indian Institute of Technology, Bombay, anshulnasery@gmail.com

## 2 Background

### 2.1 Fast Matrix Operations Using SVD

The SVD allows faster computation of many matrix operations commonly used by Neural Networks. A few examples include matrix determinant [3], matrix inverse [7], Spectral Normalization [11], the matrix exponential [8], the Cayley transform [4], weight decay, condition number and compression by low-rank approximation [16]. Proofs can be found in most linear algebra textbooks, see, e.g., [12].

### 2.2 The SVD Reparameterization

This subsection describes how [17] allows for using the SVD of the weight matrices in Neural Networks without computing them, and in particular, how this approach is limited by the computation of sequential inner products. Let $W = U\Sigma V^T$ be the SVD of a weight matrix $W$ where $\Sigma$ is a diagonal matrix and $U, V$ are orthogonal matrices, i.e, $U^T = U^{-1}$ and $V^T = V^{-1}$. The goal is to perform gradient descent updates to $W$ while preserving the SVD. Consider updating $U, \Sigma, V$ a small step $\eta \in \mathbb{R}$ in the direction of gradients $\nabla_U, \nabla_\Sigma, \nabla_V$.

$$\Sigma' = \Sigma - \eta\nabla_\Sigma, \quad U' = U - \eta\nabla_U, \quad V' = V - \eta\nabla_V.$$

While $\Sigma'$ remains diagonal, both $U'$ and $V'$ are in general not orthogonal, which is needed to preserve the SVD. To this end, [17] suggested using a technique from [10] which decomposes an orthogonal matrix as a product of $d$ Householder matrices $H_1, \ldots, H_d$:

$$U = \prod_{i=1}^{d} H_i \qquad H_i = I - 2\frac{v_i v_i^T}{||v_i||_2^2} \qquad v_i \in \mathbb{R}^d. \tag{1}$$

Householder matrices satisfy several useful properties. In particular, the matrix $U$ remains orthogonal under gradient descent updates $v_i = v_i - \eta\nabla_{v_i}$ [10]. Furthermore, all products of Householder matrices are orthogonal, and any $d \times d$ orthogonal matrix can be decomposed as a product of $d$ Householder matrices [14]. Householder matrices thus allow us to perform gradient descent over orthogonal matrices, which allows us to preserve the SVD of $W$ during gradient descent updates.

**Multiplication.** One potential issue remains. The Householder decomposition might increase the time it takes to multiply $UX$ for a mini-batch $X \in \mathbb{R}^{d \times m}$ during the forward pass. Computing $UX = H_1 \cdots (H_{d-1}(H_d \cdot X))$ takes $d$ Householder multiplications. If done sequentially, as indicated by the parenthesis, each Householder multiplication can be computed in $O(dm)$ time [17]. All $d$ multiplications can thus be done in $O(d^2m)$ time. Therefore, the Householder decomposition does not increase the time complexity of computing $UX$.

Unfortunately, the $O(d^2m)$ time complexity comes at the cost of multiplying each Householder matrix sequentially, and each Householder multiplication entails computing an inner product, see Equation (1). The multiplication $UX$ then requires the computation of $O(d)$ inner products sequentially. Such sequential computation is slow on parallel hardware like GPUs, much slower than normal matrix multiplication. To exploit GPUs, [17] suggested using a parallel algorithm that takes $O(d^3)$ time, but this is no faster than computing the SVD.

We are thus left with two options: (1) an $O(d^2m)$ sequential algorithm and (2) an $O(d^3)$ parallel algorithm. The first option is undesirable since it entails the sequential computation of $O(d)$ inner products. The second option is also undesirable since it takes $O(d^3)$ which is the same as computing the SVD, i.e., we might as-well just compute the SVD. In practice, both algorithms usually achieve no speed-up for the matrix operations like matrix inversion as we show in Section 4.2.

Our main contribution is a novel parallel algorithm, FastH, which resolves the issue with sequential inner products without increasing the time complexity. FastH takes $O(d^2m)$ time but performs $O(d/m + m)$ sequential matrix-matrix operations instead of $O(d)$ sequential vector-vector operations (inner products). In practice, FastH is up to $6.2\times$ faster than the parallel algorithm and up to $27.1\times$ faster than the sequential algorithm, see Section 4.1.

**Mathematical Setting.** We compare the different methods by counting the number of sequential matrix-matrix and vector-vector operations. We count only once when other sequential operations can be done in parallel. For example, processing $v_1, ..., v_{d/2}$ sequentially while, in parallel, processing $v_{d/2+1}, ..., v_d$ sequentially, we count $d/2$ sequential vector-vector operations.

**Orthogonal Gradient Descent.** The SVD reparameterization performs gradient descent over orthogonal matrices. This is possible with Householder matrices, however, other techniques, such as [2, 9], rely on the matrix exponential and the Cayley map, respectively. For $d \times d$ matrices both techniques spend $O(d^3)$ time, which is no faster than computing the SVD.

# 3 Fast Householder Multiplication (FastH)

## 3.1 Forward Pass

Our goal is to create an $O(d^2m)$ algorithm with few sequential operations that solves the following problem: Given an input $X \in \mathbb{R}^{d \times m}$ with $d > m > 1$ and Householder matrices $H_1, ..., H_d$, compute the output $A = H_1 \cdots H_d X$. For simplicity, we assume $m$ divides $d$.

Since each $H_i$ is a $d \times d$ matrix, it would take $O(d^3)$ time to read the input $H_1, ..., H_d$. Therefore, we represent each Householder matrix $H_i$ by its Householder vector $v_i$ such that $H_i = I - 2v_i v_i^T / ||v_i||_2^2$. A simplified version of the forward pass of FastH proceeds as follows: divide the Householder product $H_1 \cdots H_d$ into smaller products $P_1 \cdots P_{d/m}$ so each $P_i$ is a product of $m$ Householder matrices:

$$P_i = H_{(i-1) \cdot m + 1} \cdots H_{i \cdot m} \qquad i = 1, ..., d/m. \tag{2}$$

All $d/m$ products $P_i$ can be computed in parallel. The output can then be computed by $A = P_1 \cdots P_{d/m} X$ instead of $A = H_1 \cdots H_d X$, which reduces the number of sequential matrix multiplications from $d$ to $d/m$.

This algorithm computes the correct $A$. However, the time complexity increases due to two issues. First, multiplying each product $P_i$ with $X$ takes $O(d^2m)$ time, a total of $O(d^3)$ time for all $d/m$ products. Second, we need to compute all $d/m$ products $P_1, ..., P_{d/m}$ in $O(d^2m)$ time, so each product $P_i$ must be computed in $O(d^2m/(d/m)) = O(dm^2)$ time. If we only use the Householder structure, it takes $O(d^2m)$ time to compute each $P_i$, which is not fast enough.

Both issues can be resolved, yielding an $O(d^2m)$ algorithm. The key ingredient is a linear algebra result [1] that dates back to 1987. The result is restated in Lemma 1.

**Lemma 1.** *For any $m$ Householder matrices $H_1, ..., H_m$ there exists $W, Y \in \mathbb{R}^{d \times m}$ st. $I - 2WY^T = H_1 \cdots H_m$. Computing $W$ and $Y$ takes $O(dm^2)$ time and $m$ sequential Householder multiplications.*

For completeness, we provide pseudo-code in Algorithm 1. Theorem 1 states properties of Algorithm 1 and its proof clarify how Lemma 1 solves both issues outlined above.

**Theorem 1.** *Algorithm 1 computes $H_1 \cdots H_d X$ in $O(d^2m)$ time with $O(d/m + m)$ sequential matrix multiplications.*

*Proof.* **Correctness.** Each iteration of Step 2 in Algorithm 1 utilizes Lemma 1 to compute $A_i = A_{i+1} - 2W_i(Y_i^T A_{i+1}) = P_i A_{i+1}$. Therefore, at termination, $A_1 = P_1 \cdots P_{d/m} X$. In Step 1, we used Lemma 1 to compute the $P_i$'s such that $A = H_1 \cdots H_d X$ as wanted.

**Time Complexity.** Consider the for loop in Step 1. By Lemma 1, each iteration takes $O(dm^2)$ time. Therefore, the total time of the $d/m$ iterations is $O(dm^2 d/m) = O(d^2m)$. Consider iteration $i$ of the loop in Step 2. The time of the iteration is asymptotically dominated by both matrix multiplications. Since $A_{i+1}, X_i$ and $Y_i$ are $d \times m$ matrices, it takes $O(dm^2)$ time to compute both matrix multiplications. There are $d/m$ iterations so the total time becomes $O(dm^2 d/m) = O(d^2m)$.

**Number of Sequential Operations.** Each iteration in Step 2 performs two sequential matrix multiplications. There are $d/m$ sequential iterations which gives a total of $O(d/m)$ sequential matrix multiplications. Each iteration in Step 1 performs $m$ sequential Householder multiplications to construct $P_i$, see Lemma 1. Since each iteration is run in parallel, the algorithm performs no more than $O(d/m + m)$ sequential matrix multiplications. $\qquad \square$

**Remark.** Section 3.2 extends the techniques from this section to handle gradient computations. For simplicity, this section had Algorithm 1 compute only $A_1$, however, in Section 3.2 it will be convenient to assume $A_1, ..., A_{d/m}$ are precomputed. Each $A_i = P_i \cdots P_{d/m} X$ can be saved during Step 2 of Algorithm 1 without increasing asymptotic memory consumption.

## 3.2 Backwards Propagation

This subsection extends the techniques from Section 3.1 to handle gradient computations. Our goal is to create an $O(d^2 m)$ algorithm with few sequential operations that solves the following problem: Given $A_1, \ldots, A_{d/m+1}, P_1, \ldots, P_{d/m}$ and $\frac{\partial L}{\partial A_1}$ for some loss function $L$, compute $\frac{\partial L}{\partial X}$ and $\frac{\partial L}{\partial v_1}, \ldots, \frac{\partial L}{\partial v_d}$, where $v_j$ is a Householder vector st. $H_j = I - 2 v_j v_j^T / ||v_j||_2^2$.

Since each $P_i$ is a $d \times d$ matrix, it would take $O(d^3/m)$ time to read the input $P_1, \ldots, P_{d/m}$. Therefore, we represent each $P_i$ by its WY decomposition $P_i = I - 2WY^T$.

On a high-level the backward pass of FastH has two steps.

**Step 1.** Sequentially compute $\frac{\partial L}{\partial A_2}, \frac{\partial L}{\partial A_3}, \ldots, \frac{\partial L}{\partial A_{d/m+1}}$ by

$$\frac{\partial L}{\partial A_{i+1}} = \left[ \frac{\partial A_i}{\partial A_{i+1}} \right]^T \frac{\partial L}{\partial A_i} = P_i^T \frac{\partial L}{\partial A_i} \tag{3}$$

This also gives the gradient wrt. $X$ since $X = A_{d/m+1}$.

**Step 2.** Use $\frac{\partial L}{\partial A_1}, \ldots, \frac{\partial L}{\partial A_{d/m}}$ from Step 1 to compute the gradient $\frac{\partial L}{\partial v_j}$ for all $j$. This problem can be split into $d/m$ subproblems, which can be solved in parallel, one subproblem for each $\frac{\partial L}{\partial A_i}$.

**Details.** For completeness, we state pseudo-code in Algorithm 2, which we now explain with the help of Figure 2. Figure 2a depicts a computational graph of Step 1 in Algorithm 2. In the figure, consider $\frac{\partial L}{\partial A_1}$ and $P_1^T$, which both have directed edges to a multiplication node (denoted by $\cdot$). The output of this multiplication is $\frac{\partial L}{\partial A_2}$ by Equation (3). This can be repeated to obtain $\frac{\partial L}{\partial A_2}, \ldots, \frac{\partial L}{\partial A_{d/m+1}}$.

Step 2 computes the gradient of all Householder vectors $\frac{\partial L}{\partial v_j}$. This computation is split into $d/m$ distinct subproblems that can be solved in parallel. Each subproblem concerns $\frac{\partial L}{\partial A_i}$ and the product $P_i$, see line 8-10 in Algorithm 2.

To ease notation, we index the Householder matrices of $P_i$ by $P_i = \widehat{H}_1 \cdots \widehat{H}_m$. Furthermore, we let $\widehat{A}_{m+1} := A_{i+1}$ and $\widehat{A}_j := \widehat{H}_j \widehat{A}_{j+1}$. The notation implies that $\widehat{A}_1 = \widehat{H}_1 \cdots \widehat{H}_m \widehat{A}_{m+1} = P_i A_{i+1} = A_i$. The goal of each subproblem is to compute gradients wrt. the Householder vectors $\widehat{v}_m, \ldots, \widehat{v}_1$ of $\widehat{H}_m, \ldots, \widehat{H}_1$. To compute the gradient of $\widehat{v}_j$, we need $\widehat{A}_{j+1}$ and $\frac{\partial L}{\partial \widehat{A}_j}$, which can be computed by:

$$\widehat{A}_{j+1} = \widehat{H}_j^{-1} \widehat{A}_j = \widehat{H}_j^T \widehat{A}_j \qquad \frac{\partial L}{\partial \widehat{A}_{j+1}} = \left[ \frac{\partial \widehat{A}_j}{\partial \widehat{A}_{j+1}} \right]^T \frac{\partial L}{\partial \widehat{A}_j} = \widehat{H}_j^T \frac{\partial L}{\partial \widehat{A}_j} \tag{4}$$

Figure 2b depicts how $\widehat{A}_2, \ldots, \widehat{A}_{m+1}$ and $\frac{\partial L}{\partial \widehat{A}_2}, \ldots, \frac{\partial L}{\partial \widehat{A}_{m+1}}$ can be computed given $\widehat{A}_1$ and $\frac{\partial L}{\partial \widehat{A}_1}$. Given $\widehat{A}_{j+1}$ and $\frac{\partial L}{\partial \widehat{A}_j}$, we can compute $\frac{\partial L}{\partial \widehat{v}_j}$ as done in [10, 17]. For completeness, we restate the needed equation in our notation, see Equation (5).

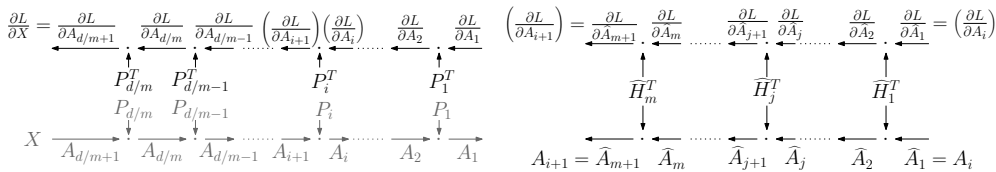

(a) Step 1: Sequential part of Algorithm 2.     (b) Step 2: The $i$'th subproblem in Algorithm 2.

Figure 2: Computational graph of Step 1 and the $i$'th subproblem in Step 2 from Algorithm 2.

Let $a^{(l)}$ be the $l$'th column of $\widehat{A}_{j+1}$ and let $g^{(l)}$ be the $l$'th column of $\frac{\partial L}{\partial \widehat{A}_j}$. The sum of the gradient over a mini-batch of size $m$ is then:

$$-\frac{2}{||\widehat{v}_j||_2^2}\sum_{l=1}^{m}(\widehat{v}_j^T a^{(l)})g^{(l)} + (\widehat{v}_j^T g^{(l)})a^{(l)} - \frac{2}{||\widehat{v}_j||_2^2}(\widehat{v}_j^T a^{(l)})(\widehat{v}_j^T g^{(l)})\widehat{v}_j \qquad (5)$$

Theorem 2 states properties of Algorithm 2.

**Theorem 2.** *Algorithm 2 computes $\frac{\partial L}{\partial X}$ and $\frac{\partial L}{\partial v_1},...,\frac{\partial L}{\partial v_d}$ in $O(d^2m)$ time with $O(d/m+m)$ sequential matrix multiplications.*

*Proof.* See the Supplementary Material 8.1. □

---

**Algorithm 1** FastH Forward

1: **Input:** $X \in \mathbb{R}^{d\times m}$ and $H_1,...,H_d \in \mathbb{R}^{d\times d}$.

2: **Output:** $A_1 = H_1\cdots H_d X$.

3: // Step 1
4: **for** $i = d/m$ **to** 1 **do in parallel**
5:     Compute $Y_i, W_i \in \mathbb{R}^{d\times m}$ st.
        $P_i = I - 2W_iY_i^T$    $\triangleright O(dm^2)$
        by using Lemma 1.
6: **end for**

7: // Step 2
8: $A_{d/m+1} = X$.
9: **for** $i = d/m$ **to** 1 **do sequentially**
10:    $A_i = A_{i+1} - 2W_i(Y_i^T A_{i+1})$. $\triangleright O(dm^2)$
11: **end for**
12: **return** $A_1$.

---

**Algorithm 2** FastH Backward

1: **Input:** $A_1,...,A_{d/m+1}, P_1,...,P_{d/m}$ and $\frac{\partial L}{\partial A_1}$.

2: **Output:** $\frac{\partial L}{\partial X}$ and $\frac{\partial L}{\partial v_k}$ for all $k$ where $H_k = I - 2\frac{v_kv_k^T}{||v_k||_2^2}$.

3: // Step 1
4: **for** $i = 1$ **to** $d/m$ **do sequentially**
5:    $\frac{\partial L}{\partial A_{i+1}} = P_i^T \frac{\partial L}{\partial A_i}$ eq. (3).    $\triangleright O(dm^2)$
6: **end for**

7: // Step 2
8: **for** $i = 1$ **to** $d/m$ **do in parallel**
9:    Let $\frac{\partial L}{\partial \widehat{A}_1} = \left(\frac{\partial L}{\partial A_i}\right)$.
10:    To ease notation, let $P_i = \widehat{H}_1\cdots\widehat{H}_m$.
11:    **for** $j = 1$ **to** $m$ **do**
12:       Compute $\widehat{A}_{j+1}, \frac{\partial L}{\partial \widehat{A}_j}$ see eq. (4).   $\triangleright O(dm)$
13:       Compute $\frac{\partial L}{\partial \widehat{v}_j}$ using $\widehat{A}_{j+1}, \frac{\partial L}{\partial \widehat{A}_j}$, eq. (5). $\triangleright O(dm)$
14:    **end for**
15: **end for**
16: **return** $\frac{\partial L}{\partial X} = \frac{\partial L}{\partial A_{d/m+1}}$ and $\frac{\partial L}{\partial v_k}$ for all $k = 1,...,d$.

---

## 3.3 Extensions

**Trade-off.** Both Algorithm 1 and Algorithm 2 can be extended to take a parameter $k$ that controls a trade-off between *total time complexity* and *the amount of parallelism*. This can be achieved by changing the number of Householder matrices in each product $P_i$ from the mini-batch size $m$ to an integer $k$. The resulting algorithms take $O(d^2k + d^2m)$ time, $O(d^2m/k)$ space and has $O(d/k+k)$ sequential matrix multiplications. This extension has the practical benefit that one can try different values of $k$ and choose the one that yields superior performance on a particular hardware setup. Note that we never need to search for $k$ more than once. The number of sequential matrix multiplications $O(d/k+k)$ is minimized when $k = O(\sqrt{d})$. For a constant $c > 1$, we can find the best $k \in \{2,3,...,c\lceil\sqrt{d}\rceil\}$ by trying all $O(\sqrt{d})$ values. The search needs to be done only once and takes $O(\sqrt{d}(d^2k + d^2m)) = O(d^3 + d^{2.5}m)$ time. In practice, the time it took to find $k$ was negligible, e.g., on the hardware we describe in Section 4 we found $k$ in less than $1s$ for $d = 784$.

**Rectangular Matrices.** We can use the SVD reparametrization for rectangular $W \in \mathbb{R}^{n\times m}$. Use orthogonal $U \in \mathbb{R}^{n\times n}, V \in \mathbb{R}^{m\times m}$ and diagonal $\Sigma \in \mathbb{R}^{n\times m}$ and let $W = U\Sigma V^T$.

**Convolutional Layers.** So far, we have considered the SVD reparameterization for matrices which corresponds to fully connected layers. The matrix case extends to convolutions by, e.g., $1 \times 1$ convolutions [7]. The SVD reparameterization can be used for such convolutions without increasing the time complexity. On an input with height $h$ and width $w$ FastH performs $O(d/m + mhw)$ sequential matrix multiplications instead of the $O(d)$ sequential inner products.

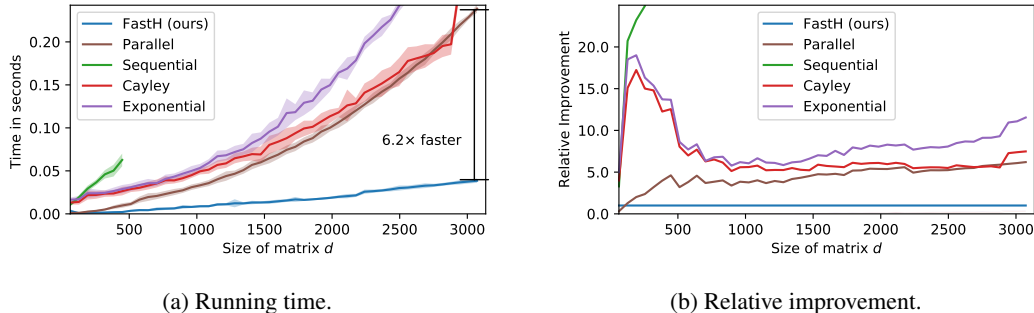

(a) Running time.                                    (b) Relative improvement.

Figure 3: Comparisons of the running times for FastH against previous algorithms. The sequential algorithm from [17] crashed when $d > 448$. (a) Running times of different algorithms for $d \times d$ matrices. (b) Running times of FastH relative to previous algorithms, i.e., the mean time of a previous algorithm divided by the mean time of FastH.

**Recurrent Layers.** The SVD reparameterization was developed for Recurrent Neural Networks (RNNs) [17]. Let $r$ be the number of recurrent applications of the RNN. FastH performs $O(d/m+rm)$ sequential matrix operations instead of the $O(d)$ sequential inner products.

## 4   Experiments

This section contains two experiments. Section 4.1 compares the running time of FastH against alternatives. Section 4.2 shows that FastH speeds up matrix operations. To simulate a realistic machine learning environment, we performed all experiments on a standard machine learning server using a single NVIDIA RTX 2080 Ti.

### 4.1   Comparing Running Time

This subsection compares the running time of FastH against four alternative algorithms. We compare the time all algorithms spend on gradient descent with a single orthogonal matrix, since such constrained gradient descent dominates the running time of the SVD reparameterization.

We first compare FastH against the parallel and sequential algorithm from [17], all three algorithms rely on the Householder decomposition. For completeness, we also compare against approaches that does not rely on the Householder decomposition, in particular, the matrix exponential and the Cayley map [2][3]. See Supplementary Material 8.2 for further details.

We measure the time of a gradient descent step with a weight matrix $W \in \mathbb{R}^{d \times d}$ and a mini-batch $X \in \mathbb{R}^{d \times m}$, where $m = 32$ and $d = 1 \cdot 64, 2 \cdot 64, ..., 48 \cdot 64$. We ran each algorithm 100 times, and we report mean time $\mu$ with error bars $[\mu - \sigma, \mu + \sigma]$ where $\sigma$ is the standard deviation of running time over the 100 repetitions.

Figure 3a depicts the running time on the y-axis, as the size of the $d \times d$ matrices increases on the x-axis. For $d > 64$, FastH is faster than all previous approaches. At $d = 64$ FastH is faster than all previous approaches, except the parallel algorithm. Previous work employ sizes $d = 192$ in [7] and $d = 784$ in [17].

Figure 3b depicts how much faster FastH is relative to the previous algorithms, i.e., the mean time of a previous algorithm divided by the time of FastH, which we refer to as relative improvement. For $d > 500$, the relative improvement of FastH increases with $d$.

At $d = 448$ FastH is roughly $25\times$ faster than the sequential algorithm. FastH is even faster with $d = 3072$ than the sequential algorithm with $d = 448$. Previous work like [6, 15] use the Householder decomposition with the sequential algorithm. Since FastH computes the same thing as the sequential algorithm, it can be used to reduce computation time with no downside.

Table 1: Relating standard method to matrix decompositions for computing matrix operations.

| Matrix Operation | Standard Method | SVD or Eigendecomposition |
|---|---|---|
| Determinant | TORCH.SLOGDET(W) | $\sum_{i=1}^{d} \lg|\Sigma_{ii}|$ |
| Inverse | TORCH.INVERSE(W) | $V\Sigma^{-1}U^T$ |
| Matrix Exponential | Padé Approximation [2] | $Ue^{\Sigma}U^T$ |
| Cayley map | TORCH.SOLVE(I-W, I+W) | $U(I{-}\Sigma)(I{+}\Sigma)^{-1}U^T$ |

## 4.2 Using the SVD to Compute Matrix Operations

This subsection investigates whether the SVD reparameterization achieves practical speed-ups for matrix operations like matrix inversion. We consider four different matrix operations. For each operation, we compare the SVD reparameterization against the standard method for computing the specific matrix operation, see Table 1.

**Timing the Operations.** The matrix operations are usually used during the forward pass of a Neural Network, which change the subsequent gradient computations. We therefore measure the sum of the time it takes to compute the matrix operation, the forward pass and the subsequent gradient computations.

For example, with matrix inversion, we measure the time it takes to compute the matrix operation $\Sigma^{-1}$, the forward pass $W^{-1}X = V\Sigma^{-1}U^T X$ and the subsequent gradient computation wrt. $U, \Sigma, V$ and $X$. The measured time is then compared with TORCH.INVERSE, i.e, we compare against the total time it takes to compute TORCH.INVERSE(W), the forward pass $W^{-1}X$, and the subsequent gradient computation wrt. $W$ and $X$.

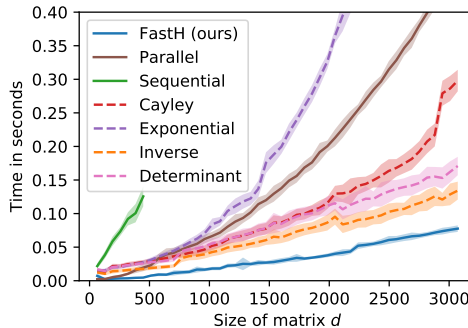

Figure 4: Running time of matrix operations. Solid lines depict approaches which use the SVD reparameterization and dashed lines depict standard methods like TORCH.INVERSE.

**Setup.** We run the SVD reparameterization with three different algorithms: FastH, the sequential and the parallel algorithm from [17]. For each matrix operation, we consider matrices $V, \Sigma, U, W \in \mathbb{R}^{d \times d}$ and $X \in \mathbb{R}^{d \times m}$, where $m = 32$ and $d = 1 \cdot 64, 2 \cdot 64, ..., 48 \cdot 64$. We repeat the experiment 100 times, and report the mean time $\mu$ with error bars $[\mu - \sigma, \mu + \sigma]$ where $\sigma$ is the standard deviation of the running times over the 100 repetitions. To avoid clutter, we plot only the time of FastH for the matrix operation it is slowest to compute, and the time of the sequential and parallel algorithms for the matrix operation they were fastest to compute.

Figure 4 depicts the measured running time on the y-axis with the size of the $d \times d$ matrices increasing on the x-axis. Each matrix operation computed by a standard method is plotted as a dashed line, and the different algorithms for the SVD reparameterization are plotted as solid lines. In all cases, FastH is faster than the standard methods. For example, with $d = 768$, FastH is $3.1\times$ faster than the Cayley map, $4.1\times$ faster than the matrix exponential, $2.7\times$ faster than inverse and $3.5\times$ faster than matrix determinant. The sequential algorithm is not fast enough to speed up any matrix operation.

## 5 Related Work

**The Householder Decomposition.** The Householder decomposition of orthogonal matrices has been used in much previous works, e.g., [6, 10, 13, 15, 17]. Previous work typically use a type of sequential algorithm that performs $O(d)$ sequential inner products. To circumvent the resulting long computation time on GPUs, previous work often suggest limiting the number of Householder matrices, which limits the expressiveness of the orthogonal matrix, introducing a trade-off between computation time and expressiveness.

FastH takes the same asymptotic time as the sequential algorithm, however, it performs less sequential matrix operations, making it up to $27\times$ faster in practice. Since FastH computes the same output as the previous sequential algorithms, it can be used in previous work without degrading the performance of their model. In particular, FastH can be used to either (1) increase expressiveness at no additional computational cost or (2) retain the same level of expresiveness at lower computational cost.

**SVDs in Neural Networks.**   The authors of [17] introduced a technique that provides access to the SVD of the weights in a Neural Network without computing the SVD. Their motivation for developing this technique was the exploding/vanishing gradient issue in RNNs. In particular, they use the SVD reparameterization to force all singular values to be within the range $[1 \pm \epsilon]$ for some small $\epsilon$.

We point out that although their technique, in theory, can be used to speed up matrix operations, their algorithms are too slow to speed-up most matrix operations in practice. To mitigate this problem, we introduce a new algorithm that is more suitable for GPUs, which allows us to speed up several matrix operations in practice.

**Different Orthogonal Parameterizations.**   The SVD reparameterization by [17] uses the Householder decomposition to perform gradient descent with orthogonal matrices. Their work was followed by [4] that raises a theoretical concern about the use of Householder decomposition. Alternative approaches based on the matrix exponential and the Cayley map have desirable provable guarantees, which currently, it is not known whether the Householder decomposition possesses. This might make it desirable to use the matrix exponential or the Cayley map together with the SVD reparameterization from [17]. However, previous work spend $O(d^3)$ time to compute or approximate the matrix exponential and the Cayley map. These approaches are therefore undesirable, because they share the $O(d^3)$ time complexity with SVD and thus cannot speed up SVD computations.

**Normalizing Flows.**   Normalizing Flows [3] is a type of generative model that, in some cases [6, 7], entails the computation of matrix determinant and matrix inversion. [7] propose to use the PLU decomposition $W = PLU$ where $P$ is a permutation matrix and $L, U$ are lower and upper triangular. The decomposition allows the determinant computation in $O(d)$ time instead of $O(d^3)$. [6] point out that a fixed permutation matrix $P$ limits flexibility. To fix this issue, they suggest using the $QR$ decomposition where $R$ is a rectangular matrix and $Q$ is orthogonal. They suggest making $Q$ orthogonal by using the Householder decomposition which FastH can speed up. Alternatively, one could use the SVD decomposition instead of the QR or PLU decomposition.

# 6   Code

To make FastH widely accessible, we wrote a PyTorch implementation of the SVD reparameterization which uses the FastH algorithm. The implementation can be used by changing just a single line of code, i.e, change NN.LINEAR to LINEARSVD. While implementing FastH, we found that Python did not provide an adequate level of parallelization. We therefore implemented FastH in CUDA to fully utilize the parallel capabilities of GPUs. Code: `github.com/AlexanderMath/fasth/`.

# 7   Conclusion

We pointed out that, in theory, the techniques from [10, 17] allow decreasing the time complexity of several matrix operations used in Neural Networks. However, in practice, we demonstrated that the techniques are not fast enough on GPUs for moderately sized use-cases. We proposed a novel algorithm, FastH, that remedies the issues with both algorithms from [17], which is up to $27\times$ faster than the previous sequential algorithm. FastH introduces no loss of quality, it computes the same result as the previous algorithms, just faster. FastH brings two immediate benefits: (1) improves upon the techniques from [17] in such a way that it is possible to speed up matrix operations, and (2) speeds up previous work that employ the Householder decomposition as done in, e.g., [6, 13, 15].

## Broader Impact

Our algorithm speeds up the use of Householder decompositions in Neural Networks. This can positively impact researchers who use Householder decompositions, since they will be able to execute experiments faster. This is particularly beneficial for researchers with a constraint on their computational budget, in other words, a PhD student with one GPU stands to benefit more than a lab with state-of-the-art GPU computing infrastructure. The reduction in computing time also decrease power consumption and thus carbon emissions. However, as a potential negative impact, it is possible that the decrease in computation time will increase the usage of Neural Networks and thus increase overall carbon emission.

**Acknowledgements** Alexander Mathiasen was supported by Kasper Green Larsen's AUFF Starting Grant. Frederik Hvilshøj was supported by DIGIT Aarhus University Centre for Digitalisation, Big Data and Data Analytics. Jakob Rødsgaard Jørgensen was supported by the Independent Research Fund Denmark. Finally, we thank DIGIT Aarhus University Centre for Digitalisation for providing us with GPU capabilities.

## Footnotes

[3]For the matrix exponential and the Cayley map we used the open-source implementation https://github.com/Lezcano/expRNN from [2]. For the Householder decomposition, we used the open-source implementation https://github.com/zhangjiong724/spectral-RNN of the sequential and parallel algorithm from [17].

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
