[Supplementary Material]

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

Consider $\frac{\partial L}{\partial X}$ computed in Step 1:

$$\frac{\partial L}{\partial X} = \frac{\partial L}{\partial A_{d/m+1}} = P_{d/m}^T \cdots P_1^T \frac{\partial L}{\partial A_1}$$

$$= H_d^T \cdots H_1^T \frac{\partial L}{\partial A_1}. \qquad\qquad eq. \ (2)$$

This is the same as that computed in [17].

Consider Step 2. Both $\frac{\partial L}{\partial \widehat{v}_j}$ and $\frac{\partial L}{\partial \widehat{A}_j}$ are computed as done in [17]. $\widehat{A}_{j+1}$ is computed using Equation (4) similar to backpropagation without storing activations [5], but using the fact that $\widehat{H}_j^T = \widehat{H}_j^{-1}$.

**Time Complexity.** In Step 1, the for loop performs $d/m$ matrix multiplications. Due to the WY decomposition $P_i^T = (I - 2WY^T)^T = I - 2YW^T$ which can be multiplied on $\frac{\partial L}{\partial A_i} \in \mathbb{R}^{d \times m}$ in $O(dm^2)$ time since $W, Y \in \mathbb{R}^{d \times m}$. The computation is repeated $d/m$ times, and take a total of $O(d^2 m)$ time.

Step 2 line 12 in Algorithm 3 performs two Householder matrix multiplications which take $O(dm)$ time, see Equation (4). In line 13, the gradient is computed by Equation (5), this sum also takes $O(dm)$ time to compute. Both computations on line 12 and 13 are repeated $d/m \cdot m$ times, see line 8 and line 11. Therefore, the total time is $O(d^2 m)$.

**Number of Sequential Operations.** Step 1 performs $O(d/m)$ sequential matrix operations. Lines 11-14 of Step 2 perform $O(m)$ sequential matrix multiplications. Since each iteration of line 8-15 is run in parallel, the algorithm performs no more than $O(d/m + m)$ sequential matrix multiplications.

$\square$

---

**Algorithm 3** FastH Backward

---

1: **Input:** $A_1, ..., A_{d/m+1}$, $P_1, ..., P_{d/m}$ and $\frac{\partial L}{\partial A_1}$.
2: **Output:** $\frac{\partial L}{\partial X}$ and $\frac{\partial L}{\partial v_k}$ for all $k$ where $H_k = I - 2 \frac{v_k v_k^T}{||v_k||_2^2}$.
3: // Step 1
4: **for** $i = 1$ **to** $d/m$ **do sequentially**
5: $\quad \frac{\partial L}{\partial A_{i+1}} = P_i^T \frac{\partial L}{\partial A_i}$ eq. (3). $\qquad\qquad\qquad\qquad\qquad\qquad\qquad\quad \triangleright O(dm^2)$
6: **end for**

7: // Step 2
8: **for** $i = 1$ **to** $d/m$ **do in parallel**
9: $\quad$ Let $\frac{\partial L}{\partial \widehat{A}_1} = \left( \frac{\partial L}{\partial A_i} \right)$.
10: $\quad$ To ease notation, let $P_i = \widehat{H}_1 \cdots \widehat{H}_m$.
11: $\quad$ **for** $j = 1$ **to** $m$ **do**
12: $\qquad$ Compute $\widehat{A}_{j+1}$, $\frac{\partial L}{\partial \widehat{A}_j}$ see eq. (4). $\qquad\qquad\qquad\qquad\qquad\quad \triangleright O(dm)$
13: $\qquad$ Compute $\frac{\partial L}{\partial \widehat{v}_j}$ using $\widehat{A}_{j+1}, \frac{\partial L}{\partial \widehat{A}_j}$, eq. (5). $\qquad\qquad\qquad\quad \triangleright O(dm)$
14: $\quad$ **end for**
15: **end for**
16: **return** $\frac{\partial L}{\partial X} = \frac{\partial L}{\partial A_{d/m+1}}$ and $\frac{\partial L}{\partial v_k}$ for all $k = 1, ..., d$.

---

## 8.2 Comparing Running Time

This subsection clarifies how the matrix exponential and the Cayley map was used in combination with the SVD reparameterization from [17]. It also provides further details on the exact computations we timed in the experiment. These details were left out of the main article as they require the introduction of some notation regarding a *reparameterization function*.

Let $V \in \mathbb{R}^{d \times d}$ be a weight matrix and let $\phi$ be a function that reparameterizes $V$ so $\phi(V)$ is orthogonal, and we can perform gradient descent wrt. $V$. The Householder decomposition can be used to construct such a function $\phi$, by letting the columns of $V$ be Householder vectors and $\phi(V)$ be the product of the resulting Householder matrices.

There exist alternative ways of constructing $\phi$ which does not rely on the Householder decomposition. For example, the matrix exponential approach where $\phi_{exp}(V) = e^V$ and the Cayley map approach where $\phi_C(V) = (I - V)(I + V)^{-1}$ [2].

We record the joint time it takes to compute $\phi(V)X$ and the gradients wrt. $V$ and $X$ for a dummy input $X \in \mathbb{R}^{d \times M}$. To simplify the gradient computation of $V$, we use a dummy gradient $G \in \mathbb{R}^{d \times M}$ st. the gradient wrt. $V$ is $[\frac{\partial \phi(V) \cdot X}{\partial V}]^T G$. It might be useful to think of $G$ as the gradient that arises by back-propagating through a Neural Network.

Both the dummy input and the dummy gradient have normally distributed entries $X_{ij}, G_{ij} \sim N(0, 1)$.

**Implementation Details.** The parallel algorithm from [17] halted for larger values of $d$. The failing code was not part of the main computation. This allowed us to remove the failing code and still get a good estimate of the running time of the parallel algorithm. We emphasize that removing the corresponding code makes the parallel algorithm faster. The experiments thus demonstrated that FastH is faster than a lower bound on the running time of the parallel algorithm.

## 8.3 Using the SVD to Compute Matrix Operations

This section requires first reading Section 4.1 and Section 4.2. Recall that we, in Section 4.2, want to measure the total time it takes to compute both the matrix operation, the forward pass and the gradient computations. For example, with matrix inversion, we want to compute the matrix operation $\Sigma^{-1}$, the forward pass $V\Sigma^{-1}U^T X$ and the gradient computations wrt $V, \Sigma, U, X$.

The time of the forward pass and gradient computations is no more than two multiplications and two gradient computations, which is exactly two times what we measured in Section 4.1. We re-used those measurements, and add the time it takes to compute the matrix operation, e.g., $\Sigma^{-1}$.

**Over Estimating the Time of FastH.** The matrix exponential and the Cayley map require one orthogonal matrix instead of two, i.e., $U\Sigma U^T$ instead of $U\Sigma V^T$. The WY decomposition then only needs to be computed for $U$ and not both $U$ and $V$. By re-using the data, we measure the time of two orthogonal matrices, this thus estimates an upper-bound of the real running time of FastH.