[Reviews · NeurIPS 2020]

Review 1

Summary and Contributions: The paper proposes a novel way to accelerate the multiplication of Householder matrices, and demonstrates substantial speedup over prior work [18] and the standard Pytorch implementation. The speedup could benefit researchers who use SVD in neural networks for matrix inversion, exponential, spectral normalization, etc.

Strengths: The problem of speeding up SVD in neural networks is quite novel. I admit that I do not know much about this community to evaluate its size, but the results in this paper show substantial impact to this community. The empirical evaluation is sound. I did not have time to go through the theoretical grounding.

Weaknesses: Although the maximum speedups over [18] in matrix inversion is quite sizable (27x in Fig 1), the gain over the much simpler, standard baseline seems moderate (2.7x to 4.3x). This potentially weakens the significance of the contributions, unless the proposed approach has advantages in other dimensions compared to the baseline.

Correctness: The claims, method and empirical methodology is correct.

Clarity: Yes. The paper clearly explains the motivation, problem, and solution of the work.

Relation to Prior Work: Yes. The authors made is clear that previous work on SVD reparameterizations are O(d^3) while they focus on getting a O(d^2m) algorithm.

Reproducibility: Yes

Additional Feedback:


Review 2

Summary and Contributions: This paper considers how to efficiently incorporate SVD-decomposed dense layer in a DNN. The core idea of representing the U and V matrix as a product of rank-1 Householder reflections was already present in [18]. This allows efficient forward and backward propagation, i.e. no computational overhead. In return, many operations now become free. However, the benefit is "on paper", since the method presented in [18] is not amenable to parallelization. The present paper solves this issue by breaking the product of Householder reflection into blocks, and converting each block to its WY representation. The WY representation is a way to represent a product of Householder reflections that was invented to solve similar issues in parallel multifrontal QR decomposition algorithms.

Strengths: The paper a practical problem in a previous theoretical advancement. It was already observed that SVD layers can be useful in deep learning, and it was shown that this can be done efficiently, however in practice it didn't work well due to lack of parallelism. This paper solves this. The techniques in the paper can have applicability beyond deep learning. The authors release code (w/ CUDA).

Weaknesses: 1) Novelty: At the end, there is very little in the way of a fundamentally new idea. The use of SVD in deep learning, and the observation that it can be represented efficiently via Householder reflections, was already developed in [18]. WY factorization was already developed in 1987 also, I think in order to solve similar problems. The authors simply adjust the tool to the job. 2) Limited applicability: seems that the technique applies only to layers whose #input neurons is equal to #output neurons. 3) Experiments: very weak. They do not explore deep learning at all, but rather focus on the time to do a single step and the cost of various matrix operations. I am curious to see how end-to-end learning pans out. 4) Discussion of memory complexity is a bit neglected. There are some discussions throughout, but I do not feel they are complete. Is there a memory-parallelization tradeoff? The start of section 3.3 discusses only time complexity and parallelism tradeoff.

Correctness: Claims seem correct. The empirical methodology is a bit problematic since it does not explore deep learning at all, but rather on the time to do a single step and the cost of various matrix operations. I am curious to see how end-to-end learning pans out.

Clarity: Mostly. Section 3.3 is very unclear, especially how the SVD layers can be used in convolutional layers and recurrent layers. Also: Lines 158-160 are unclear. What is g? Sum of what?

Relation to Prior Work: Mostly. However, the use of WY decomposition in multifrontal QR factorization is neglected. Since WY was invented to solve similar issues in such algorithms, it should be discussed.

Reproducibility: Yes

Additional Feedback: - The mix between using v to represent the HH vectors and using U and V for the SVD components can create a challenge in understanding the paper. - Line 82 "Mathematical Setting": I think "Computational Model" is more appropriate. - Line 158-160: unclear. ,What is g? Sum of what? - Page 5, pseudocodes: Should say that H_j is given using its HH vector. - Why is the inverse needed in deep learning. - Line 275 "These approaches are thus undesirable for SVD since..." - What do you mean here? - End of line 175: What took less than 1s?


Review 3

Summary and Contributions: The paper propose a method to incorporate SVD of Deep Neural Network layers in training in an efficient way, which can have applications in Recurrent Neural Networks. The authors identify that previous attempts incurred many serialized inner product operations that resulted in higher time complexity and reduce resource utilization in massively parallel GPUs. They propose a method named Fast Householder Multiplication which uses an algorithm to compute Householder matrices with low time complexity. They analytically show that FastH can reduce the number of serialized computations and reduced the time complexity. They further experimentally compare their method with previous algorithms based on serialized inner products and show that they can outperform these methods.

Strengths: Due to the rapid development of Machine Learning hardware during the past few years, each of which employ various techniques for purposes of acceleration, theoretical guarantees of performance, such as the ones derived in the paper, have become valuable. The authors have further included their implementation that can help reproduce their results.

Weaknesses: However, I would like to ask the authors to provide more details regarding the experimental setup to help reproducibility.

Correctness: The analytical derivations seem correct. Further, the experimental results provided seem to follow the adhere the theoretical bounds derived in the paper.

Clarity: The paper is well written and clear.

Relation to Prior Work: Yes, the authors have clearly discussed the previous approaches for incorporating SVD into neural networks and provided the necessary details for algebraic results used upon which they have based their proposed methodology.

Reproducibility: Yes

Additional Feedback:


Review 4

Summary and Contributions: This paper proposes to accelerate the matrix operations in practice. The new algorithm convert the vector-vector operations to matrix-matrix operations to improve the degree of parallelism. A time complexity analysis is provided on matrix multiplications and gradient computations. Besides, this paper provides two algorithms for Forward and Backward.

Strengths: The strengths of this work include proposing a new method for handle matrix operations with high speed, a theoretical analysis on the complexity of Forward Pass and Backwards Propagation.

Weaknesses: First, something is confusing about why should we take into account the computation of sequential inner products. I'm not sure about it without any theoretical or empirical analysis. Second, the proposed method is rather incremental upon previous works. The new algorithm is based on the grouped matrix operations. Third, there is a gap between theoretical time-complexity and empirical time-complexity which makes the analysis of time-complexity in section 3 can't support the effectiveness of the designed algorithm. Based on the result in section 4.2, the sequential method spend too much time on inner products isn't mentioned in section 3. Besides, in section 3.3, the cost of searching proper k is O(d^3) which is 'negligible' which makes me confuse about which is not negligible in designed algorithm.

Correctness: The adopted methodology seems to be correct. The empirical results seems to be convinced.

Clarity: Satisfactory.

Relation to Prior Work: Yes.

Reproducibility: Yes

Additional Feedback:

[Author Response · NeurIPS 2020]

**R1, 7/10** *Although the maximum speedups over [18] in matrix inversion is quite sizable (27x in Fig 1), the gain over*
*the much simpler, standard baseline seems moderate (2.7x to 4.3x). This potentially weakens the significance of the*
*contributions, unless the proposed approach has advantages in other dimensions compared to the baseline.*

Indeed, the approach has advantages in other dimensions. FastH attains 5x to 27x speed-up for orthogonal matrices as
used by [4,6,7,8,10,14,15,16,18], i.e, we can expect 5x to 27x if we use FastH in the referenced approaches (Figure 3).
In particular, we found FastH to be so much faster than previous methods, that it can speed up even matrix inversion,
something that was not possible with previous methods. We updated the introduction to reflect this distinction.

**R4, 5/10** *The cost of searching proper $k$ is $O(d^3)$ which is 'negligible' which makes me confused about which is not*
*negligible in designed algorithm.*

Thanks for raising this concern. The default $k = batch\_size$ used in our experiments works well and one does not need
to search for $k$. However, one might be able to improve slightly by trying different $k$. This only needs to be done **once**
for a given hardware setup. This differs to our algorithm, FastH, which is used **every step during training**, i.e., FastH
is used $10^5$ times while the search is done 1 time. We rephrased a few sentences in section 3.3 to clarify this confusion.

*The result in section 4.2, the sequential method spend too much time on inner products isn't mentioned in section 3.*

We opted to use Section 3 to introduce our FastH algorithm, while clarifying the mentioned issue in the introduction
(L22-28) and the background section (L66-70). We believe the current organization allows for a sharper separation
between previous work and our proposed algorithm FastH.

*There is a gap between theoretical time-complexity and empirical time-complexity which makes the analysis of time-*
*complexity in section 3 can't support the effectiveness of the designed algorithm.*

We believe there is a misconception here. "FastH retains the **same desirable time complexity** as the sequential
algorithm from [18] while reducing the number of sequential operations" (introduction L31-33). In other words, FastH
is $27x$ faster than [18] due to less sequential work, **not** due to a difference in time complexity.

**R2, 6/10** *Limited applicability: seems the technique applies only to layers whose #input neurons = #output neurons.*

Thanks for raising this concern. Our technique does apply when the number of input neurons $n$ is different to number of
output neurons $m$, that is, for a regular linear layer with weight matrix $W \in \mathbb{R}^{n \times m}$. The weight matrix has a singular
value decomposition $W = U\Sigma V^T$ for orthogonal $U, V$ where $U \in \mathbb{R}^{n \times n}$, $V \in \mathbb{R}^{m \times m}$ and diagonal $\Sigma \in \mathbb{R}^{n \times m}$.
FastH works for both $U$ and $V$. This can furthermore be extended to attain semi-orthogonal $W$. [1] We added a paragraph
in the subsection concerning extensions with the hopes that it clarifies this confusion.

*The empirical methodology is a bit problematic since it does not explore deep learning at all, but rather on the time to*
*do a single step and the cost of various matrix operations.*

Thanks for raising this concern. The use of Householder matrices in deep learning has received much attention in
previous work, e.g., [6,10,14,16,18]. FastH computes **exactly the same** as the algorithm used by [6,10,14,16,18];
repeating their experiments with FastH would thus attain the same results, albeit faster. Since we believe [6,10,14,16,18]
adequately demonstrate the usefulness of Householder matrices for deep learning, we found the additional value of
more such experiments were not that high. Furthermore, there are additional benefits to studying the performance of
single operators as opposed to end-to-end deep learning experiments. Time complexity is a more transparent measure
to investigate than the validation loss of deep learning models. Such measure shows the benefits of our approach
**irrespective** of the architecture, optimizer, loss function and the many hyperparameters of complex networks.

*Novelty: There is very little in the way of a fundamentally new idea. ... . The authors simply adjust the tool to the job.*

As evidence against the claimed lack of novelty, we present three articles that would benefit from a $O(dm^2)$ parallel
algorithm but did not "simply adjust the tool to the job." [18] realized the issue with sequential computation and
suggested a parallel $O(d^3)$ algorithm. [10] uses the related CWY decomposition for gradient computations, without
realizing WY can increase GPU utilization. They instead speed up by using fewer Householder matrices. An
article contemporaneous[2] to our submission (`https://arxiv.org/abs/2004.08675`) addresses parallel Householder
products with the related CWY decomposition, but attains a $O(d^3)$ bound (see their table 2, serial complexity is $L^3$).

**R3, 7/10.** *I would like to ask the authors to provide more details regarding the experimental setup to help reproducibility.*

We will soon open-source "neuralsvd.py" from the supplementary material, which we updated to run our main experi-
ment and draw Figure 3. We also updated "README.txt" to contain more details regarding the experimental setup.

of which the authors are righteously not aware.

## Footnotes

[1]Semi-orthogonal means $W^T W = I \neq W W^T$. This is true if $n > m$ and $\Sigma_{ii} = 1$.

[2]Contemporaneous, as per NeurIPS guidelines, refers to a work published less than two months before the submission deadline


[Meta-Review · NeurIPS 2020]

This paper carefully applies known linear algebra results to represent the SVD of weight matrices in neural networks, allowing efficient forward and backward passes in parallel computing environments. The analysis is correct and the experiments convincingly demonstrate speedup by the proposed method. The author also open-sourced their efficient implementation of their algorithm, which can be quite useful for the community and can inspire/accelerate future research.